# Large Questionnaire Survey on Sleep Duration and Insomnia Using the TV Hybridcast System by Japan Broadcasting Corporation (NHK)

**DOI:** 10.3390/ijerph18052691

**Published:** 2021-03-07

**Authors:** Kazuki Ito, Hiroshi Kadotani, Isa Okajima, Ayaka Ubara, Mamoru Ichikawa, Chie Omichi, Towa Miyamoto, Arichika Matsuda, Yukiyoshi Sumi, Hirotoshi Kitagawa

**Affiliations:** 1Department of Anesthesiology, Shiga University of Medical Science, Shiga 520-2192, Japan; momo3@belle.shiga-med.ac.jp (K.I.); hirotosi@belle.shiga-med.ac.jp (H.K.); 2Department of Sleep and Behavioral Sciences, Shiga University of Medical Science, Shiga 520-2192, Japan; okajima-i@tokyo-kasei.ac.jp (I.O.); cykc1005@mail2.doshisha.ac.jp (A.U.); comichi@koto.kpu-m.ac.jp (C.O.); arichika@belle.shiga-med.ac.jp (A.M.); 3Department of Psychological Counseling, Faculty of Humanities, Tokyo Kasei University, Tokyo 173-8602, Japan; 4Graduate School of Psychology, Doshisha University, Kyoto 610-0394, Japan; 5JSPS Research Fellowships, Tokyo 102-0083, Japan; 6Japan Broadcasting Corporation, Tokyo 150-8001, Japan; mamoruichikawa@gmail.com; 7Department of Psychiatry, Graduate School of Medical Science, Kyoto Prefectural University of Medicine, Kyoto 602-8566, Japan; 8Department of Psychiatry, Shiga University of Medical Science, Shiga 520-2192, Japan; twmymt@belle.shiga-med.ac.jp (T.M.); elasticvisco@gmail.com (Y.S.)

**Keywords:** insomnia, sleep duration, internet survey, hybridcast, bidirectional network, TV program, participation bias

## Abstract

Background: Japanese people are known to have the shortest sleep duration in the world. To date, no study has assessed a large Japanese population for insomnia and sleep duration. Methods: We performed an Ιnternet-based survey in association with a national television (TV) program. Questionnaire data were collected not only through personal computers, tablets, and smartphones, but also through the Hybridcast system, which combines broadcasts over airwaves with broadband data provided via the Internet using the TV remote controller. The Athens Insomnia Scale (AIS) was used to assess insomnia. Results: A total of 301,241 subjects participated in the survey. Participants slept for an average of 5.96 ± 1.13 h; the average AIS score was 6.82 ± 3.69. A total of 26.1% of male and 27.1% of female participants had both insomnia (AIS ≥ 6) and short sleep duration (<6 h). Responses were recorded through the Hybridcast system for 76.4% of the elderly (age ≥ 65 years) subjects and through personal computers, tablets, or smartphones for 59.9–82.7% of the younger subjects (age ≤ 65 years). Conclusions: Almost a quarter of the Japanese participants presented short sleep duration and insomnia. Furthermore, the Hybridcast system may be useful for performing large internet-based surveys, especially for elderly individuals.

## 1. Introduction

Sleep debt represents an accumulation of sleep deprivation. Along with causing sleepiness, it deteriorates daily work performance, cognitive function, physical and mental health, and causes serious accidents [1,2,3,4]. It is reported that insomnia and short sleep duration are associated with mortality and medical morbidities, such as diabetes, hypertension, and depression [5,6]. In addition, the importance of combining insomnia and sleep duration has been pointed out, and insomniac people with short sleep duration have an increased risk of these medical morbidities, as compared to people with insomnia or with short sleep duration alone [6]. A total of 21.4% of the Japanese population is reported to have insomnia [7]. The average sleep duration has been found to be approximately 7.5 h globally [8]; while it is reported to be about 7 h among the Japanese population, which is one of the shortest in the world [8,9,10]. A total of 36.1% and 42.1% of Japanese men and women, respectively, sleep for less than 6 h [11]. Sleep duration among Japanese women is shorter than that in men [11]. Sleep duration in both sexes is the shortest in the middle-aged than in the other age groups [11]. The sleep duration of the Japanese population has reduced over the last few decades; in the 1970s, the sleep duration was almost 8 h, while in 2015, on weekdays, the population slept for only an average of 7 h and 15 min [12]. The adverse effects of sleep debt or insufficient sleep are not only limited to individuals but also affect society as a whole. Insufficient sleep can result in a large economic loss in terms of decreased gross domestic product (GDP). It is estimated that the loss of economic cost in Japan due to insufficient sleep or sleep debt is about USD 138 billion, which is about 2.9% of the GDP [13]. Although insomnia and sleep debt are major problems associated with medical and mental health in Japan, no survey on both insomnia and sleep duration on a large Japanese population has been conducted.

We performed a large web-based survey (*n* > 300,000) for sleep duration and insomnia in association with a national television (TV) program [14]. Large-scale conventional questionnaire surveys through mail or face-to-face interviews can be time- and cost-consuming; however, web-based surveys could possibly overcome these major limitations. On the contrary, the use of devices, such as personal computers (PCs), tablets, and smartphones, requires certain skills, and is not suitable for web-based surveys among elderly individuals. Elderly people may not be familiar with these devices, and they still use TV as an information source medium [15]. Nowadays, TVs are not only devices for receiving broadcast radio waves, but they can also serve as an Internet device. The use of a new TV service, called ‘Hybridcast TV’, which combines broadcasting and the Internet, has been increasing over the last decade [16,17]. NHK (Nippon Hoso Kyokai: Japan Broadcasting Corporation) Hybridcast broadcasting was first launched by NHK on 2 September 2013. This technology makes it possible for viewers to respond to questionnaires with their TV remote controllers during television broadcast, helping researchers aggregate data using the Internet [18]. The method for using it is easy, such that children in the lower grades of elementary school are able to use it as well. However, as far as we know, no researchers have used the Hybridcast TV system to perform a large-scale national questionnaire survey in Japan. We performed a web-based questionnaire survey in association with a TV program using the Hybridcast TV system, PCs, tablets, and smartphones [14].

In this study, we aimed to analyse the following: (1) the prevalence of insomnia and short sleep duration in a large Japanese population and (2) the age- and sex-based differences in choosing an Internet device to respond to a questionnaire attached to a TV program.

## 2. Materials and Methods

### 2.1. Participants

NHK had broadcast a special TV program on sleep debt (insufficient sleep) on 18 June 2017 [14]. The participants were TV program viewers or NHK website visitors who responded to a questionnaire survey attached to the TV program.

The NHK website started advertising the special TV program on sleep debt on the Internet and requested prospective participants to answer certain questions on insomnia. The program was broadcast on TV from 21:00 to 21:59 and the data of responses for sleep duration and insomnia were aggregated using the Hybridcast system, tablets, computers, and smartphones [14].

We explained the purpose of this study through the TV program and the NHK homepage [14], and regarded electronic participation of subjects as their informed consent. All data were collected automatically and anonymously. This study was approved by the Ethics Committee of Shiga University of Medical Science (29-013).

### 2.2. Data Collection Dates/Methods

Data collection dates and methods were as follows:

(a) The NHK website started advertising the special TV program on sleep debt on the Internet from 12:00 AM on 11 June 2017 [14]. The announcements through the website were continued even after program completion.

(b) Data collection through the Internet using tablets, computers, and smartphones was performed from 12:00 AM on 11 June 2017 to 12:00 AM on 26 June 2017 (prior to, during, and after airing of the program).

(c) The program was first broadcasted on TV from 21:00 to 21:59 on 18 June 2017. The program was rebroadcasted on-demand through the Internet thereafter [14].

(d) Data collection using the Hybridcast system was performed only during the first broadcast (from 21:00 to 21:59 on 18 June 2017).

In total, 33.1%, 50.4%, and 16.5% of the participants answered the survey prior to, during, and after the airing of the program, respectively.

### 2.3. Systems Used for the Survey

We conducted a cross-sectional questionnaire survey using the Internet. The participants responded to the survey through PCs, tablets, or smartphones, as well as through the Hybridcast TV system, which was modified for undertaking the survey.

A Hybridcast service combines programs broadcast over airwaves with broadband data provided via the Internet to impart a variety of services (Figure 1). In our survey, during the broadcast of the TV program, the TV anchor announced and introduced the start of the Hybridcast system. Questions were provided one by one by the TV anchor. Participants started responding to questions when requested, by pressing the ‘d button’ (data button) on the remote controller to start the Hybridcast system (Figure 2). Questions provided either had responses as a selection of one item out of four answering items or as a selection of one item from a pull-down list. Participants could select one of the four-coloured buttons, followed by pressing the ‘enter’ button to answer the four-choice questions. The other questions could be answered by selecting an item from the pull-down list by using the ‘up’ and ‘down’ arrows (left and right to the ‘enter’ button) and, then, pressing the ‘enter’ button. Participants could answer through the Hybridcast system during the broadcast of the TV program. Participants could also respond through their own PCs, tablets, or smartphones at any time after the broadcast of the TV program [18].

Solid and dashed arrows indicate “d button” (data button) used to start the Hybridcast system and ‘enter’ button, respectively. Blue, red, yellow, and green colour buttons (white circle) can be used to answer four-choice questions.

### 2.4. Questionnaires

The participants were enquired about their sex (male or female), age (<20, 20–39, 40–64, ≥65 years) and usual sleep duration in 30-min intervals, and the Athens Insomnia Scale (AIS) questionnaire was administered. In Japan, those aged <20 year and ≥20 years are considered minors and adults, respectively. Those between the ages of 40 and 64 years are considered middle-aged, and those over 65 years old are considered elderly. We thought that four-choice questions were easier to answer than pull-down questions and would encourage active participation. Thus, we chose to use a four-choice question with responses corresponding to age ranges of <20, 20–39, 40–64, and ≥65 years in this survey.

The AIS is an eight-item self-reporting questionnaire that measures the intensity of sleep difficulties according to the International Statistical Classification of Disease and Related Health Problems 10th Revision (ICD-10) diagnostic criteria for insomnia [19]. The total scores range from 0 to 24 [19,20]. The AIS follows eight items: AIS_1: sleep initiation; AIS_2: awakening during night; AIS_3: early morning awakening; AIS_4: total sleep duration; AIS_5: overall quality of life; AIS_6: problems with sense of well-being; AIS_7: overall functioning; and AIS_8: daytime sleepiness. Each AIS item is rated on a four-point scale (0 = no problem at all, 1 = slightly problematic, 2 = markedly problematic, and 3 = extremely problematic). An AIS score of 6 is widely used as a cut-off value to detect insomnia [20]. AIS total scores of 6–9, 10–15, and 16–24 are classified as mild, moderate, and severe insomnia, respectively [21].

### 2.5. Statistical Analysis

Statistical analysis was performed using IBM SPSS Statistics 25.0 (IBM, Armonk, NY, USA). Referring to previous studies [6], we used the cut-off values of AIS (<6 or ≥6 points) and sleep duration (<6 h or ≥6 h) to classify the patients into four categories: insomnia with short sleep duration (AIS ≥ 6 points and sleep duration < 6 h), insomnia alone (AIS ≥ 6 points and sleep duration ≥6 h), short sleep duration alone (AIS < 6 points and sleep duration < 6 h), and normal sleep (AIS < 6 points and sleep duration ≥ 6 h).

Continuous and categorical variables were evaluated with an unpaired one-way analysis of variance (ANOVA) or a chi-square test, respectively. The effect size of one-way ANOVA was presented as η^2^. The effect size was considered small, medium, and large when the η^2^ value reached 0.01, 0.06, and 0.14, respectively [22]. The association of nominal or ordinal data with more than two categories of unequal columns and rows in the data matrix was explored using Cramer’s V, which follows the principles of chi square statistics with a similar interpretation of value between 0 and 1. The association and the effect size were considered small, medium, and large when Cramer’s V value reached 0.1, 0.3, and 0.5, respectively [22].

## 3. Results

In total, 301,783 subjects participated in the survey. Data for 542 subjects were insufficient, and those of the remaining 301,241 subjects were analysed in this study. The effects of sex on participants’ age, sleep duration, and AIS score were almost negligible with Cramer’s V values < 0.1 (Table 1).

Sleep duration was the shortest in the 40 to 64-year-old group for both sexes. Sleep duration was longest in the <20-year-old group for both sexes (Table 2). Age had a small effect on sleep duration with η^2^ = 0.027. Effects of sex and generation on sleep duration were almost negligible with η^2^ < 0.01.

Almost negligible effects were found in insomnia/sleep duration and age in both sexes with Cramer’s V values < 0.1 (Table 3). Normal sleep (AIS < 6 points and sleep duration ≥ 6 h) was decreased in people in the 20- to 64-year-old group in both sexes. About 15.1–29.8% of the participants of the same generation had insomnia and short sleep duration. A total of 26.1% (36484/139880) male and 27.1% (43792/161361) female participants had insomnia with short sleep duration. In addition, the proportion of insomnia alone (39.5% male and 40.8% female) was higher in the elderly (≥65 years), but the proportion of short sleep duration alone (13.1% male and 14.0% female) or insomnia with short sleep duration (29.0% male and 29.8% female) was higher in the middle age (40–64 years) group.

Sex and age had a small effect on the type of device used for answering the questionnaire with Cramer’s V values > 0.1 (Table 4). Among male participants, the use of the Hybridcast system was higher and the use of smartphones was lower compared to that among female participants. A total of 76.4% of participants who were aged ≥65 years responded to the survey using the Hybridcast system. Participants from the younger age group used smartphones more frequently; especially, 64.3% of the participants in the 20–39-year age group used smartphones.

The questionnaire data were collected between 12:00 AM on 11 June 2017 and 12:00 AM on 26 June 2017. Almost half (50.4%) of the survey responses were received during the broadcast of the TV program (from 21:00 to 21:59 on 18 June 2016) (Table 5). Cramer’s V values of time of recording responses for sleep duration and the AIS score categories were 0.099 and 0.067, respectively (Appendix A). Effects of sex, sleep duration, and insomnia symptoms at the time of recording responses were almost negligible; a small effect was associated with age, and a large effect was associated with the type of device used for responding to the survey. Participants aged ≥65 years tended to respond during the broadcast of the TV program. The Hybridcast system could be used for responding to the questionnaire only during the broadcast of the program, while other devices could be used at any time.

## 4. Discussion

We performed a large web-based survey (*n* > 300,000) for sleep duration and insomnia in association with a national TV program [14].

Average sleep duration was the shortest in the 40 to 64-year age group (Table 2). This is also true in the Japanese general population, according to the National Health and Nutrition Survey 2017 in Japan, where sleep duration was found to be the shortest among the 40 to 49-year age group, for both sexes [11].

Prevalence of participants with insomnia (AIS ≥ 6) and short sleep duration (<6 h) was the highest in 40–64-year-old participants for both sexes (Table 3). Approximately 26.1% of male and 27.1% of female participants had insomnia and short sleep duration; it was especially common in the working age group of 40–64 years. This is the first study to examine the proportions of the four categories of insomnia and sleep debt by age and sex. It has been reported that insomnia with short sleep duration (<6 h) is associated with mortality and medical morbidities, such as deficits in neuropsychological performance, type 2 diabetes, and hypertension [6]. More than a quarter of our participants had increased risks for morbidities. Therefore, it was necessary to encourage them to go to the hospital and get active evidence-based treatment, such as hypnotics and cognitive behavioural therapy [23]. In the TV program and the Web-based survey, participants with AIS scores of 6–9 and ≥10 were designated to have moderate and high risks of sleep problems, respectively. We encouraged participants with morbidity risks to improve sleep quality through lifestyle changes, which were suggested in the TV program. We expected those who failed to have improvements in sleep quality would go to the hospital.

Participants of this survey had higher AIS score and shorter sleep duration compared with other populations in Japan. AIS score and sleep duration in this population were 6.82 ± 3.69 and 5.96 ± 1.13 h, respectively (Table 1). AIS scores among city government employees and patients in a sleep apnoea outpatient unit in Japan were 4.98 ± 3.57 [4] and 3.64 ± 3.15 [24], respectively. Sleep duration in Japanese general population was reported to be 7.25–7.37 h [8,12]. These differences may have resulted from participation bias. This survey was attached to a TV program on sleep debt [14]. Audiences, especially those with enough interest to respond to the survey, might be more likely to have sleep problems including sleep debt. Survey methods might also affect participation behaviours and results. Internet survey results were reported to have large discrepancies from those of the official national census of Japan [25].

The survey was open for anyone who could use a PC, tablet, or smartphone even after the TV program was broadcasted. Half of the participants responded during the broadcast of the TV program (Table 5). This might suggest that TV still has substantial effects as a mass media device compared to other Internet services in Japan.

The main audience of the NHK TV program was the elderly group. The audience rating of the TV program (on air on 21:00–21:59 on 18 June 2017) attached to this survey has not been published, but audience ratings of two other TV programs by NHK that were broadcast on the evening of 18 June 2017 were published. They had audience ratings of 0–5%, 7–10%, 15–16%, and 21–24% for individuals aged 7–49, 50–59, 60–69, and ≥ 70 years, respectively [26]. The main audience of the TV program thought to be associated with this survey may have been the elderly. However, middle-aged individuals were the main participants in this study. A total of 9.28% and 60.37% of the participants were aged ≥65 and 40–64 years, respectively (Table 1). This discrepancy may be attributed to differences in the use of the Internet according to age.

The responses were recorded through the Hybridcast system and smartphones by 76.4% and 6.8% of elderly population (≥65 years), respectively (Table 4). More than half of younger population (<40 years) responded through smartphones. It is reported that elderly individuals do not use the Internet, especially through their smartphones, as much as younger individuals. Japanese individuals aged 13–59, 60–69, 70–79, and ≥80 years present Internet usage rates of 92.4–98.7%, 73.9%, 46.7%, and 20.1%, respectively, according to the Ministry of Internal Affairs and Communications [27]. Similarly, Japanese individuals aged 20–59, 60–69, 70–79 and ≥80 years, have smartphone usage rates of 65.5–68.8%, 47.4%, 27.2%, and 8.1%, respectively, according to official statistic sources [27]. In the future, smartphones may be the best devices for responding to Internet surveys. The response rate for Internet surveys is usually low among elderly individuals, as performing Internet surveys is considered a challenging task for them [28]. Japanese individuals aged 20–29, 30–39, 40–49, 50–59, and ≥65 years have Hybridcast use rates of 25.0%, 32.4%, 34.2%, 26.8%, and 22.5%, respectively [29]. Hybridcast usage was not age-dependent, whereas Internet or smartphone use is. The Hybridcast system may be useful for performing Internet surveys among elderly individuals.

This study had some limitations. The participants were audiences of a TV program on sleep debts and visitors of the Web page for the TV program. These audiences may have an active interest in sleep debt as they may already have sleep issues. The TV program was first broadcasted in the evening, and data collection using the Hybridcast system was performed only during the first broadcast. The elderly population with morning chronotype may have difficulty in participating in the survey. We asked individuals to categorize their age into categories (<20, 20–39, 40–64 and ≥65 years) instead of asking the actual age of the participants during data collection. We failed to analyse more detailed association with age in this study. Our results probably differ from those of the general population. Furthermore, sleep duration was not calculated by objective measurements, such as polysomnography or actigraphy.

## 5. Conclusions

A large Internet survey can be performed in association with a national TV program. Almost a quarter of this Japanese population may have short sleep duration and insomnia, which are risk factors for mortality and medical morbidities, such as diabetes, hypertension, and depression. The Hybridcast system may be useful for performing a large Internet survey, especially in elderly individuals.

## Figures and Tables

**Figure 1 ijerph-18-02691-f001:**
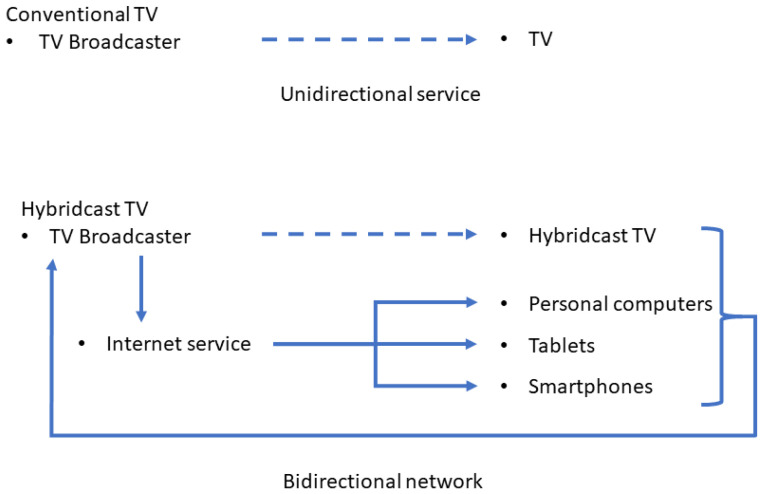
Overview of conventional TV and Hybridcast TV. Solid lines indicate bidirectional interaction between television (TV) station and audiences through the Internet. Dashed lines indicate broadcasts from the TV station to audiences over airwaves.

**Figure 2 ijerph-18-02691-f002:**
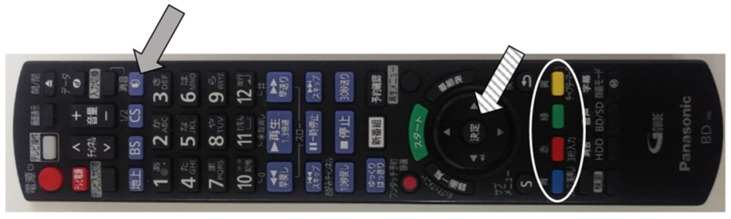
Television (TV) remote controller used in the Hybridcast TV system.

**Table 1 ijerph-18-02691-t001:** Demographic data of participants.

Variables	Total, *n* (%)		Male, *n* (%)		Female, *n* (%)		Cramer’s V
Sex	301,241	(100.0%)	139,880	(100.0%)	161,361	(100.0%)	
Age (years):							
<20	13,762	(4.57%)	6112	(4.37%)	7650	(4.74%)	0.043
20–39	77,650	(25.8%)	34,549	(24.7%)	43,101	(26.7%)	
40–64	181,881	(60.4%)	84,537	(60.4%)	97,344	(60.3%)	
≥65	27,948	(9.28%)	14,682	(10.5%)	13,266	(8.22%)	
Sleep duration (h):	5.96 ± 1.13		5.96 ± 1.16		5.95 ± 1.11		
<5	29,881	(9.92%)	13,914	(9.95%)	15,967	(9.90%)	0.018
5–6	84,566	(28.1%)	38,155	(27.3%)	46,411	(28.8%)	
6–7	123,515	(41.0%)	58,253	(41.6%)	65,262	(40.4%)	
7–8	46,967	(15.6%)	22,089	(15.8%)	24,878	(15.4%)	
8–9	12,772	(4.24%)	5855	(4.19%)	6917	(4.29%	
≥9	3540	(1.18%)	1614	(1.15%)	1926	(1.19%)	
AIS:	6.82 ± 3.69		6.80 ± 3.75		6.83 ± 3.64		
0–5	122,025	(40.5%)	57,137	(40.9%)	64,888	(40.2%)	0.010
6–9	115,679	(38.4%)	53,130	(38.0%)	62,549	38.76%	
10–15	56,619	(18.8%)	26,276	(18.8%)	30,343	18.80%	
16–24	6918	(2.30%)	3337	(2.39%)	3581	2.22%	

Mean ± SD are presented for sleep duration and the AIS score. Cramer’s V was calculated to present the effect size of the association between sex and age, sleep duration, and the AIS score among participants. Cramer’s V values ≥ 0.1 suggest small effect. AIS: Athens insomnia scale.

**Table 2 ijerph-18-02691-t002:** Average sleep duration (h) by age groups.

Age (Years)	<20	20–39	40–64	≥65	Overall
Total	6.53 ± 1.85	6.10 ± 1.19	5.82 ± 1.01	6.14 ± 1.08	5.96 ± 1.13
Male	6.74 ± 2.03	6.03 ± 1.18	5.83 ± 1.02	6.22 ± 1.16	5.96 ± 1.16
Female	6.36 ± 1.68	6.15 ± 1.19	5.81 ± 1.00	6.04 ± 0.98	5.95 ± 1.10

Sleep durations are presented as means ± standard deviations. Participants were divided into four generations as follows: <20, 20–39, 40–64, and ≥65 years old. η^2^ < 0.01; The η^2^ values for sex, age group, and sex age group were 0.001, 0.027, and 0.003, respectively.

**Table 3 ijerph-18-02691-t003:** Prevalence of insomnia/sleep duration according to the age groups.

Age Goup	Normal Sleep, *n* (%)	Insomnia Alone, *n* (%)	Short Sleep Duration Alone, *n* (%)	Insomnia with Short Sleep Duration, *n* (%)	Cramer’s V
Male					
<20	2716 (44.4%)	2102 (34.4%)	374 (6.12%)	920 (15.1%)	0.092
20–39	9913 (28.7%)	12,625 (36.5%)	3315 (9.60%)	8696 (25.2%)	
40–64	23,210 (27.5%)	25,733 (30.4%)	11,062 (13.1%)	24,532 (29.0%)	
≥65	5713 (38.9%)	5799 (39.5%)	834 (5.68%)	2336 (15.9%)	
Female					
<20	2647 (34.6%)	2603 (34.0%)	529 (6.92%)	1871 (24.5%)	0.087
20–39	12,474 (28.9%)	16,537 (38.4%)	3506 (8.13%)	10,584 (24.6%)	
40–64	26,548 (27.3%)	28,130 (28.9%)	13,658 (14.0%)	29,008 (29.8%)	
≥65	4633 (34.9%)	5411 (40.8%)	893 (6.73%)	2329 (17.6%)	

Participants were divided into four generations as follows: <20, 20–39, 40–64, and ≥65 years old. Sleep of the participants was divided into four generations as follows: insomnia with short sleep duration (AIS ≥ 6 points and sleep duration < 6 h), insomnia alone (AIS ≥ 6 points and sleep duration ≥ 6 h), short sleep duration alone (AIS < 6 points and sleep duration < 6 h), and normal sleep (AIS < 6 points and sleep duration ≥ 6 h). Cramer’s V values ≥ 0.1 suggested a small effect. Numbers presented on the columns represent the number of participants and percentages in categories. Insomnia is defined as an AIS score ≥ 6.

**Table 4 ijerph-18-02691-t004:** Types of Internet device used to answer the questionnaire survey.

Sex/Age	Hybridcast, *n* (%)	PC, *n* (%)	Tablet, *n* (%)	Smartphone, *n* (%)	Cramer’s V
Sex:					
Male	59,687 (42.7%)	26,745 (19.1%)	5320 (3.8%)	48,128 (34.4%)	0.164
Female	53,104 (32.9%)	20,489 (12.7%)	7443 (4.6%)	80,325 (49.8%)	
Age, years:					
<20	4996 (36.3%)	819 (6.0%)	534 (3.9%)	7413 (53.9%)	0.211
20–39	13,472 (17.3%)	12,222 (15.7%)	1994 (2.6%)	49,962 (64.3%)	
40–64	72,973 (40.1%)	30,461 (16.7%)	9263 (5.1%)	69,184 (38.0%)	
≥65	21,350 (76.4%)	3732 (13.4%)	972 (3.5%)	1894 (6.8%)	

Cramer’s V was calculated to present the effect size of the association between sex, age, and device use among the participants. Cramer’s V values ≥ 0.1 suggested a small effect. PC, personal computer.

**Table 5 ijerph-18-02691-t005:** Comparison of recording time of responses and demographic characteristics.

Demographics	During Broadcast, *n* (%)	Not at the Time of Broadcast, *n* (%)	Cramer’s V
Total	149,360 (49.6%)	151,881 (50.4%)	
Sex			
Male	63,810 (42.7%)	76,070 (50.1%)	0.074
Female	85,550 (57.3%)	75,811 (49.9%)	
Age (years)			
<20	7848 (5.25%)	5914 (3.89%)	0.257
20–39	26,543 (17.8%)	51,107 (33.7%)	
40–64	94,536 (63.3%)	87,345 (57.5%)	
≥65	22,954 (15.4%)	4994 (3.29%)	
Answering device			
Hybridcast	112,791 (75.5%)	0 (0.00%)	0.769
PC	7602 (5.09%)	39,632 (26.1%)	
Tablet	3615 (2.42%)	9148 (6.02%)	
Smartphone	27,873 (18.7%)	100,580 (66.2%)	

Cramer’s V was calculated to present the effect size of the association between time of recording responses and sex, age, and type of device used for responding. Cramer’s V values ≥ 0.1 suggested a small effect. PC, personal computer.

## Data Availability

The datasets analysed in the current study are available from the corresponding author upon reasonable request.

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
