# Peer review of "Large Questionnaire Survey on Sleep Duration and Insomnia Using the TV Hybridcast System by Japan Broadcasting Corporation (NHK)"

_ijerph, 2021, doi:10.3390/ijerph18052691_

Round 1

Reviewer 1 Report

Dear Authors,

This is a very interesting article which has examined sleep duration and insomnia at a national level.  The methods of data collection are particularly novel, and I feel that you have interpreted the results well.  You have acknowledged the potential limitations of participation bias from viewers that may have an interest due to their own personal circumstances, and also acknowledged the discrepancies in the data collection method.

My only comment for consideration is to possibly give more detail or discuss if the timing of the programme may have influenced the results.  You have explained that TV programme aired 21:00-22:00, would individuals with an early chronotype have chosen not to view the programme?  Was the programme repeated at a later date, or available to view on the internet at a later date which was convenient?

Author Response

The authors would like to thank the reviewer for their constructive critique that helped to improve the manuscript. We have made every effort to address the issues raised and to respond to all comments. Please, find next a detailed, point-by-point response to the reviewer's comments.

My only comment for consideration is to possibly give more detail or discuss if the timing of the programme may have influenced the results.  You have explained that TV programme aired 21:00-22:00, would individuals with an early chronotype have chosen not to view the programme?  Was the programme repeated at a later date, or available to view on the internet at a later date which was convenient?

We would like to thank the reviewer for the comment and for evaluating our manuscript. Please note that we have added information regarding on-demand rebroadcast through Internet as follows: ‘The program was first broadcasted on TV from 21:00 to 21:59 on June 18th, 2017. The program was rebroadcasted on-demand through Internet thereafter [14]’. (Lines 111–112)

Moreover, we have discussed the difficulty in participation for elderly population with morning chronotype, as a limitation. Especially, we have added the following sentence to the revised manuscript: ‘Elderly population with morning chronotype might have had difficulty in participating in the survey’. (Lines 313–314)

Reviewer 2 Report

Thank you for the opportunity to review this interesting study.  The manuscript is overall well written and offers a novel approach to the study of sleep.  I recommend publication following the authors addressing the following:

1) Offer background to justify the use of age and and sex-based research aims.

2) Offer justification for the age-based thresholds (<20, 20 - 39, etc.).  All literature in discussion includes 10-year, not 20-year threshholds.  Why twenty-year breakdowns?  A statement in limitations is needed to address this.

3) Clarify data collection dates/method as follows: a) indicate NHK on-air advertising of program before and after airing of program - when and how often prior to airing, b) clarify that  internet-based survey options (non TV remote) were open to collect responses June 11 prior to airing of program, c) clarify proportion of responses collectected prior to program airing, during program airing, and following program airing.

4) Lines 233 - 235, clarify who as well as how or in which ways people were identified and encouraged to go to the hospital.  

5) The authors may wish to (not required from this review) note that Hybridcast TV with NHK could offer a useful tool for sleep health education as a public health approach to improving sleep - perhaps targeting the elderly given their disproportioinal use of NHK.  A study likely worthy of grant funding.

Author Response

The authors would like to thank the reviewer for the careful assessment of our manuscript. We have made every effort to address the issues raised and to respond to all comments. Please, find next a detailed, point-by-point response to the reviewer's comments.

1) Offer background to justify the use of age and sex-based research aims.

Please note that we have added the background information to the revised manuscript to explain the use of age and sex-based research aims. Especially, we have added the following: ‘Sleep duration in both sexes is the shortest in the middle-aged group than in the other age groups’. (Lines 52–53)

2) Offer justification for the age-based thresholds (<20, 20 - 39, etc.).  All literature in discussion includes 10-year, not 20-year thresholds.  Why twenty-year breakdowns?  A statement in limitations is needed to address this.

We would like to thank the reviewer for the comment. Please note that we have stated the reasons why we chose four age categories, as follows: ‘The participants were enquired about their sex (male or female), age (< 20, 20–39, 40-64, ≥ 65 years) and usual sleep duration in 30-minute intervals, and the Athens Insomnia Scale (AIS) questionnaire was administered. In Japan, those aged < 20 year and > 20 years are considered minors and adults, respectively. Those between the ages of 40 and 64 years are considered middle-aged, and those over 65 years old are considered elderly. We thought that four-choice questions were easier to answer than pull-down questions and would encourage active participation. Thus, we chose to use a four-choice question with responses corresponding to age ranges of < 20, 20–39, 40–64, and ≥ 65 years in this survey.’ (Lines 148–156)

Moreover, we have discussed this issue as a limitation in the revised manuscript (Lines 314–316).

3) Clarify data collection dates/method as follows: a) indicate NHK on-air advertising of program before and after airing of program - when and how often prior to airing, b) clarify that internet-based survey options (non TV remote) were open to collect responses June 11 prior to airing of program, c) clarify proportion of responses collected prior to program airing, during program airing, and following program airing.

We would like to thank the reviewer for the comment. Please note that we have added a new subsection named ‘Data collection dates/methods’ to clearly explain the data collection dates and methods (Lines 103–117)

4) Lines 233 - 235, clarify who as well as how or in which ways people were identified and encouraged to go to the hospital. 

We would like to thank the reviewer for the comment. Please note that we have added the following text to present this information: ‘Therefore, it was necessary to encourage them to go to the hospital and get active evidence-based treatment, such as hypnotics and cognitive behavioural therapy [23]. In the TV program and the Web-based survey, participants with AIS scores of 6–9 and ≥ 10 were designated to have moderate and high risks of sleep problems, respectively. We encouraged participants with morbidity risks to improve sleep quality through lifestyle changes, which were suggested in the TV program. We expected those who failed to have improvements in sleep quality would go to the hospital.’ (Lines 260–267)

5) The authors may wish to (not required from this review) note that Hybridcast TV with NHK could offer a useful tool for sleep health education as a public health approach to improving sleep - perhaps targeting the elderly given their disproportional use of NHK.  A study likely worthy of grant funding.

We would like to thank the reviewer for the positive evaluation of our work. This encourages us to pursuit such research projects.
